# Racial Differences in the Detection Rate of Bladder Cancer Using Blue Light Cystoscopy: Insights from a Multicenter Registry

**DOI:** 10.3390/cancers16071268

**Published:** 2024-03-24

**Authors:** Seyedeh-Sanam Ladi-Seyedian, Alireza Ghoreifi, Badrinath Konety, Kamal Pohar, Jeffrey M. Holzbeierlein, John Taylor, Max Kates, Brian Willard, Jennifer M. Taylor, Joseph C. Liao, Hristos Z. Kaimakliotis, Sima P. Porten, Gary D. Steinberg, Mark D. Tyson, Yair Lotan, Siamak Daneshmand

**Affiliations:** 1Department of Urology, Norris Cancer Center, University of Southern California, Los Angeles, CA 90033, USA; sanam.ladi@vumc.org (S.-S.L.-S.); alireza.ghoreifi@med.usc.edu (A.G.); 2Allina Health Cancer Institute, Minneapolis, MN 55407, USA; badrinath.konety@allina.com; 3Department of Urology, Ohio State University, Columbus, OH 43210, USA; kamal.pohar@osumc.edu; 4Department of Urology, University of Kansas, Kansas City, KS 66045, USA; jholzbeierlein@kumc.edu (J.M.H.); jtaylor27@kumc.edu (J.T.); 5The James Buchanan Brady Urological Institute, Johns Hopkins Medical Institutions, Baltimore, MD 21224, USA; mkates@jhmi.edu; 6Lexington Medical Center, Lexington, SC 29169, USA; tbwillard@lexhealth.org; 7Michael E. DeBakey VA Medical Center, Baylor College of Medicine, Houston, TX 77030, USA; jennifer.taylor@bcm.edu; 8VA Palo Alto Health Care System, Palo Alto, CA 94304, USA; jliao@stanford.edu; 9Department of Urology, Indiana University School of Medicine, Indianapolis, IN 46202, USA; hkaimakl@iupui.edu; 10Department of Urology, University of California San Francisco, San Francisco, CA 94115, USA; sima.porten@ucsf.edu; 11Department of Urology, Allina Health Cancer Institute, University of Minnesota, Minneapolis, MN 55407, USA; gary.d.steinberg@gmail.com; 12Department of Urology, Mayo Clinic Hospital, Phoenix, AZ 85054, USA; tyson.mark@mayo.edu; 13UT Southwestern Medical Center, Dallas, TX 75390, USA; yair.lotan@utsouthwestern.edu

**Keywords:** bladder cancer, blue light cystoscopy, racial groups, transurethral resection of bladder

## Abstract

**Simple Summary:**

Blue light cystoscopy (BLC) is a technique used to find bladder tumors more effectively. However, its performance across different races remains uncertain. In this study, we looked at how well BLC detects cancer in people of different races. We collected data from patients who had bladder tumor surgeries from 2014 to 2021. Overall, we found that BLC was better at spotting tumors compared to traditional white light cystoscopy (WLC). This was true for most races, especially for Caucasian and Asian patients. When BLC was added to WLC, it increased the detection of cancer by 10%, with the biggest improvement seen in Asian patients. Interestingly, Asian patients had the highest chance of BLC correctly identifying cancer, while Hispanic patients had the best chance of ruling out cancer. This suggests that combining BLC with WLC can improve the detection of bladder cancer irrespective of race.

**Abstract:**

The use of blue light cystoscopy (BLC) has been shown to improve bladder tumor detection. However, data demonstrating the efficacy of BLC across different races are limited. Herein, we aim to evaluate heterogeneity in the characteristics of BLC for the detection of malignant lesions among various races. Clinicopathologic information was collected from patients enrolled in the multi-institutional Cysview^®^ registry (2014–2021) who underwent transurethral resection or biopsy of bladder tumors. Outcome variables included sensitivity and negative and positive predictive values of BLC and white light cystoscopy (WLC) for the detection of malignant lesions among various races. Overall, 2379 separate lesions/tumors were identified from 1292 patients, of whom 1095 (85%) were Caucasian, 96 (7%) were African American, 51 (4%) were Asian, and 50 (4%) were Hispanic. The sensitivity of BLC was higher than that of WLC in the total cohort, as well as in the Caucasian and Asian subgroups. The addition of BLC to WLC increased the detection rate by 10% for any malignant lesion in the total cohort, with the greatest increase in Asian patients (18%). Additionally, the positive predictive value of BLC was highest in Asian patients (94%), while Hispanic patients had the highest negative predictive value (86%). Our study showed that regardless of race, BLC increases the detection of bladder cancer when combined with WLC.

## 1. Introduction

An estimated 82,000 cases of bladder cancer were diagnosed in the United States in 2023, which comprises 4.2% of all cancer diagnoses. This ranks bladder cancer as the sixth most common cancer diagnosis in the United States [1]. While the majority of patients present with non-muscle-invasive bladder cancer (NMIBC), recurrence rates remain high even at the lowest grade and stage, placing patients at increased risk of progression to muscle-invasive bladder cancer (MIBC) [2]. 

The current standard for the diagnosis of bladder cancer is white light cystoscopy (WLC) and urine cytology [3]. The main limitation of WLC is the difficulty in identifying all areas of malignancy due to the multifocal nature of bladder cancer and the presence of often inconspicuous but significant lesions, such as carcinoma in situ (CIS) [4]. Newer technologies, such as blue light cystoscopy (BLC) using hexaminolevulinate (HAL/Cysview/Hexvix), have improved the management of bladder cancer [5]. This technology is dependent upon the preferential accumulation and fluorescence of photoactive porphyrin in neoplastic tissue that is visible when exposed to blue light at a certain wavelength [5]. There is a strong body of evidence demonstrating improved detection, reduced recurrence, and possibly reduced progression of NMIBC when BLC is used in conjunction with WLC [6,7,8]. Hence, HAL has been approved and used extensively in Europe and the United States for the detection of NMIBC.

Bladder cancer clinical trials have historically lacked representation from racial and ethnic minorities, thereby hindering the comprehensive understanding of treatment efficacy and its applicability to diverse patient populations [9]. Similarly, most of clinical trials on BLC were conducted in Europe or North America, and there are insufficient data to demonstrate the equivalent efficacy of BLC across different racial groups. The aim of this study is to evaluate heterogeneity in the characteristics of BLC for the detection of malignant lesions among various racial groups, using a large multicenter cohort. 

## 2. Materials and Methods

### 2.1. Study Population 

Using the Cysview^®^ prospective registry database, which captures data from 13 different US institutions, all patients who underwent transurethral resection of bladder tumor (TURBT) using both WLC and BLC between April 2014 and February 2021 were used as the source population. Patients with suspected or known NMIBC based on a prior cystoscopy or imaging, those undergoing repeat resection for restaging or recurrence, and patients who had positive urine cytology but no apparent lesion were included in this study. Any patient with porphyria, known hypersensitivity to HAL or aminolevulinate derivatives, and those with pure upper tract or prostatic urethral lesions were excluded from the registry. 

### 2.2. Procedure

Details of the procedure have been previously described by the authors [5]. Briefly, HAL was instilled via an indwelling catheter 1 to 3 h before planned TURBT. WLC and BLC were performed using a KARL STORZ D-Light C Photodynamic Diagnostic (PDD) system. The procedure began with surveying the entire bladder with WLC and then repeating the procedure under blue light. Abnormalities in the bladder mucosa during BLC were characterized by the detection of red, homogenous fluorescence.

### 2.3. Study Parameters and Outcome Measures

Study variables included baseline clinicodemographic information (race/ethnicity, age, gender, smoking) and tumor-related variables (tumor stage and grade at diagnosis). Any severe dysplasia, CIS, or T1–4 bladder cancer was considered a positive result of pathology for malignancy.

Based on race/ethnicity, patients were divided into four groups comprising Hispanic, Asian, Black/African American (AA), and White/Caucasian (non-Hispanic) patients. The independent variable was the final pathology. The primary outcome was sensitivity of BLC, WLC, and the combination of both BLC and WLC for the detection of any malignant lesion reported on the final pathology in the total cohort and each racial population. Secondary outcomes included the positive predictive value (PPV) and negative predictive value (NPV) of BLC, WLC, and both, as well as adverse events following HAL instillations in the same groups. In addition, a subgroup analysis was performed to assess BLC accuracy in detecting CIS vs. benign lesions across racial groups. The PPV of BLC was defined as the probability that a positive lesion under blue light was truly malignant, and the NPV was defined as the probability that a blue-light-negative lesion truly did not have cancer. 

### 2.4. Statistical Analysis 

Continuous variables were described using mean ± standard deviation (SD), and categorical variables were summarized by their frequency count (percentages). Fisher’s exact test was used to assess differences among various races. Comparison between BLC and WLC sensitivity across racial groups was performed using conditional logistic regression. Analyses were performed using IBM SPSS statistics version 21 (IBM Corp., Armonk, NY, USA). All *p*-values reported were 2-sided and *p* < 0.05 was considered statistically significant.

## 3. Results

### 3.1. Baseline Clinical Features 

A total of 1292 patients (2379 lesions) with complete information in the database were included in this study. Among these, 1095 (85%) were White/Caucasian (non-Hispanic), 96 (7%) were Black/AA, 51 (4%) were Asian, and 50 (4%) were Hispanic. The median (IQR) age was 71 (35–93) years and 84% of patients were men. The demographic details of each race/ethnicity are shown in Table 1. 

### 3.2. Sensitivity 

The sensitivity of WLC, BLC, and the combination of both for any malignant lesion in the total cohort was 89%, 96%, and 99%, respectively (*p* < 0.001). The sensitivity of BLC was significantly higher than that of WLC in the White/Caucasian (non-Hispanic) (96% vs. 89%, *p* < 0.001) and Asian populations (96% vs. 78%, *p* < 0.001). Nevertheless, there was no significant difference in the sensitivity of BLC compared to WLC in the Black/AA (97% vs. 93%, *p* = 0.13) and Hispanic populations (95% vs. 95%, *p* = 0.99) (Table 2). In all races, the sensitivity of BLC was significantly higher than that of WLC for the detection of CIS (*p* < 0.001) (Figure 1). 

The addition of BLC to standard WLC increased the detection rate by 10% for any malignant lesion in the total cohort. This increase was more pronounced in Asian patients (18%) and was least pronounced in Hispanic patients (2%). In the conditional logistic regression analysis for paired data, the difference in the odds ratio of BLC vs. WLC sensitivity among races was not statistically significant (*p* = 0.35) (Table 3). 

### 3.3. PPV and NPV 

We found a PPV of 82% and an NPV of 70% for BLC in the total lesions. The PPV of BLC was highest in the Asian population (94%), while the Hispanic patients had the highest NPV for BLC (86%) (Table 2). 

### 3.4. CIS Lesions

The prevalence of CIS lesions (both isolated and concomitant) was highest among the Asian population at 26.5%, followed by White/Caucasian (non-Hispanic) individuals at 20% and Black/AA patients at 19%. Notably, the Hispanic population exhibited a lower incidence at 12%. Notably, the Hispanic group had the highest rate of benign lesions (41%) and Asian patients had the lowest rate (19%). In the subgroup analysis focusing on CIS lesions, the sensitivity of BLC in detecting CIS compared to benign lesions was 95% for Black/AA patients, 94% for Asian patients, 93% for Caucasian patients, and 80% for Hispanic patients. However, the Hispanic population had the highest NPV of BLC in detecting CIS versus benign lesions (Table 4).

### 3.5. Adverse Events

Minor adverse events, including bladder pain and urinary tract pain, were reported in 17 patients following the instillation of HAL. None of these events were definitively attributed to HAL. All 17 patients were of White/Caucasian (non-Hispanic) ethnicity, with no complications observed in patients of other races.

## 4. Discussion

Several prior randomized controlled studies have confirmed the increased detection rate as well as decreased recurrence rate of NMIBC using BLC [10,11,12,13]. However, there is no study to compare the efficacy of BLC among various races. In this study, we showed that BLC has high sensitivity in the detection of bladder cancer in all races, although its difference with WLC is more prominent in the Caucasian and Asian populations. Additionally, the sensitivity of BLC was significantly higher than that of WLC for the detection of CIS in all races, which is crucial in the management of multifocal disease or potential upstaging according to the American Urological Association (AUA) risk category [3]. Furthermore, the findings of this study confirm that the use of HAL is very safe in all races with no adverse reactions.

Incorporating BLC into WLC has been shown to enhance the visualization of bladder tumors during TURBT. In a large randomized controlled trial by Stenzl et al., it was shown that in 16% of patients with Ta/T1 bladder cancer, at least one of the tumors was only detected by BLC [14]. Similar findings were reported by Grossman et al., who showed that additional lesions could be found by BLC at any stage of bladder cancer over WLC [15]. This is in line with the findings of our study, which show a significantly higher sensitivity of BLC and the combination of BLC and WLC over WLC alone in detecting bladder lesions. This was more prominent in a subgroup of patients with CIS, who are difficult to identify through WLC. Similarly, some studies reported a more than 40% increase in CIS detection with BLC [4]. These findings have influenced the guidelines to recommend the use of BLC (if available) for the detection of bladder tumors, particularly CIS [3,16]. It is important to highlight that the results of our study should be interpreted cautiously given the unequal distribution of CIS and benign lesions among various racial groups. 

There is also emerging evidence suggesting less tumor progression using BLC [7,17]. In a previous study from our group, using the Cysview Registry, we demonstrated that in the white-light-negative group, use of BLC resulted in the detection of additional lesions in 25% of patients [5]. Hermann et al. similarly found that WLC failed to detect residual tumors in 49% of patients that were identified by BLC [13]. Hence, BLC facilitates achieving a complete resection by detecting invisible lesions under white light and delineating lesion borders, thereby enhancing the efficacy of adjuvant treatments such as intravesical therapy. This will eventually translate into decreased tumor progression and recurrence. Systematic reviews and meta-analyses have demonstrated improved recurrence-free survival with BLC at 1 and 2 years [10]. This is attributed to the lower rate of progression and prolonged time to progression observed in patients undergoing BLC [7]. Apart from this, a prior study from our group revealed an upward change in AUA risk category as a result of BLC findings in 14% of patients. This led to a change in the recommended management, with 8% of cystectomies being performed due to upstaging or AUA risk migration, prompted by lesions found particularly with BLC [5]. This has been confirmed in other prospective multicenter trials, demonstrating that 17% of patients receive more appropriate treatment due to BLC [8].

Despite the high level of evidence supporting the use of BLC in the diagnosis and surveillance of patients with NMIBC, there are no studies that specifically report the race-specific performance of tumor detection using BLC. Most studies on the outcomes of bladder cancer among racial groups have focused on Caucasian and Black/AA populations, leaving little available data for Hispanic and Asian patients. A study by Wang et al. using the Surveillance, Epidemiology, and End Results (SEER) database revealed significant racial differences in terms of patients’ characteristics, clinical–pathological features, incidence, and survival. Overall, they found that Caucasian patients have the highest incidence rate of bladder cancer, followed by Black/AA, Hispanic, and Asian patients. While the survival advantage was observed in White/Caucasian (non-Hispanic) individuals, the Black/AA population had the worst survival outcome [18]. Similar findings were observed in the study by Sung et al., using the California Cancer Registry [19]. Different lifestyles could account for the racial difference in bladder cancer incidence and outcomes. Nevertheless, disparities in access to healthcare and the quality of care provided to minority populations could also impact the early diagnosis and difference in oncological outcomes [20]. BLC can play an important role in optimizing the management of bladder cancer among all racial groups, particularly minorities. 

Our findings confirm the superiority of BLC over WLC in detecting CIS among all racial groups. The same findings were observed for all bladder lesions in White/Caucasian (non-Hispanic) and Asian populations, but not in Black/AA and Hispanic populations, which may be attributed to the smaller number of these racial populations in our registry. It is worth mentioning that prior studies have shown lower odds of guideline-based treatment among minority populations with bladder cancer [21]. Implementing BLC in these racial groups, which is recommended by the guidelines and supported by the findings of our study, could potentially improve the oncological outcomes among these patients. One factor that impacts the PPV and NPV of BLC is the prevalence of disease in the population. Overall, the rate of CIS in the bladder cancer population is around 10% [22]. However, it can be observed that in our cohort, the rate of CIS was significantly higher, with 20% of Caucasian patients, 19% of AA patients, 26% of Asian patients, and 12% of Hispanic patients affected. This suggests a bias towards performing biopsy in patients with prior CIS and possibly towards preferentially using BLC, since BLC has consistently been shown to increase the detection of CIS. Interestingly, the rate of CIS is more than twice as high in Asian than in Hispanic patients, which can explain the improved PPV in the Asian population, as BLC’s highest benefit is in these patients. There was also a significant difference in the rates of benign lesions that were biopsied based on race. For example, Hispanic patients had 41% benign lesions versus 19% for Asian individuals. Notably, the greater number of benign lesions in a group can result in a higher NPV. Whether there was a geographic or racial bias in deciding when to biopsy a patient based on a lesion is unclear, but the differences among groups could impact the PPV and NPV in this study. 

Only 15% of the registered patients in our cohort are non-White/Caucasian (non-Hispanic), which is consistent with the lower representation of racial and ethnic minorities in bladder cancer studies and clinical trials [9]. A systematic review investigating racial disparities among bladder cancer clinical trials revealed a small proportion of enrollment from Black/AA (2–8%) and Hispanic (2–5%) patients [23]. However, the Cysview Registry includes a considerable number of minorities, particularly Asian individuals, compared to available studies in the literature. This may reflect the real-world population and facilitate the generalizability of our findings. 

The main limitations of this study are the inherent biases commonly found in registry databases, such as the absence of randomization and the potential for observation and selection biases (e.g., unequal distribution of CIS and benign lesions). Also, as the data were collected from multiple institutions, surgeon experiences and variations in TURBT quality could serve as potential confounders. Additionally, the registry currently lacks long-term follow-up data, limiting our ability to assess recurrence and survival rates. Nevertheless, the significant advantage of the Cysview Registry is its multicenter, high-volume, prospectively collected data, which is likely to provide accurate reflection of the detailed care provided at tertiary referral centers for patients with bladder cancer.

## 5. Conclusions

BLC increases the detection of bladder cancer when combined with WLC in all races, particularly among White/Caucasian (non-Hispanic) and Asian patients. Additionally, the sensitivity of BLC is significantly higher than that of WLC for the detection of CIS in all racial groups. Further research is warranted to elucidate the etiology of this observation, which may ultimately alter the interpretation of lesions detected by BLC.

## Figures and Tables

**Figure 1 cancers-16-01268-f001:**
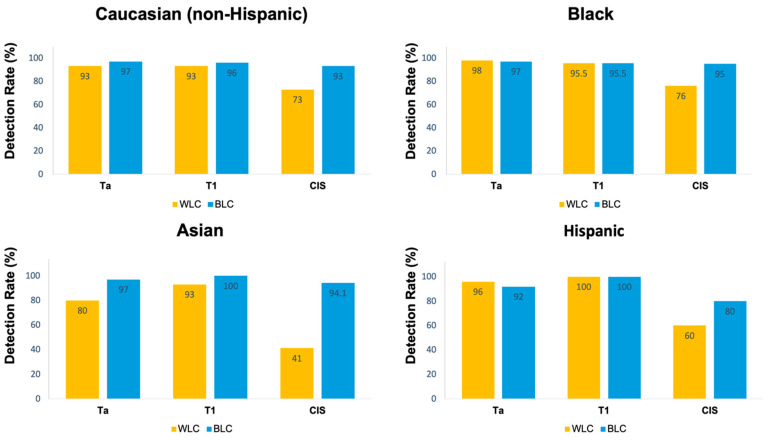
Detection rates stratified by pathology and race.

**Table 1 cancers-16-01268-t001:** Clinical and pathological characteristics of patients classified by race/ethnicity.

	Caucasian (Non-Hispanic)	African American	Asian	Hispanic
Age (mean ± SD), year	70.2 ± 9.4	67.5 ± 8.6	73.8 ± 6.3	72.2 ± 9.5
Gender (Male%)	83	80	78	84.5
Smoker (%)	67	70	57	64
Primary occurrence (%)	34	40	50	53
**Pathological stage (%)**				
Ta	60	61	50	62
T1	20	20	23.5	26
CIS *	20	19	26.5	12
**Pathological grade (%)**				
Benign	24.5	28	19	41
PUNLMP	0.5	4	0	1
Low grade	22	20	6	18
High grade	53	48	80	40

SD: standard deviation; CIS: carcinoma in situ; PUNLMP: papillary urothelial neoplasm of low malignant potential. * CIS lesions include both CIS alone and concomitant CIS.

**Table 2 cancers-16-01268-t002:** Detection rate and positive and negative predictive values (%) of different bladder lesions using white light and blue light cystoscopy.

Race (Number of Lesions)	Sensitivity			DR of BLC		Increase in DR by BLC	PPV for BLC	NPV for BLC
	WLC Only	BLC Only	Either WLC or BLC	HG	CIS			
Caucasian (non-Hispanic) (2011)	87	96	99	94	93	11	82	69
African American (178)	93	97	99	97	95	6	77	60
Asian (99)	78	96	97	94	94	18	94	60
Hispanic (91)	95	95	97	97	80	2	88	86
Total (2379)	89	96	99	94	93	10	82	70

WLC: white light cystoscopy; BLC: blue light cystoscopy; DR: detection rate; HG: high grade; CIS: carcinoma in situ; PPV: positive predictive value; NPV: negative predictive value.

**Table 3 cancers-16-01268-t003:** Odds ratio (95% CI) of comparison between BLC and WLC sensitivity, based on conditional logistic regression analysis.

Race	Odds Ratio	95% CI	*p*-Value (Score Test) #
Caucasian (non-Hispanic)	3.0	2.2–4.1	<0.001
African American	2.7	0.7–10.1	0.13
Asian	16.0	2.1–120.6	<0.001
Hispanic	1.0	0.06–16.0	1.0

# equivalent to McNemar test. *p*-value for interaction between race and method of cystoscopy (BLC vs. WLC): 0.35.

**Table 4 cancers-16-01268-t004:** Blue light accuracy to detect CIS vs. benign lesions in different races.

Race	Sensitivity	Specificity	PPV	NPV
Caucasian (non-Hispanic)	93	21	53	76
African American	95	11	39	80
Asian	94	40	73	80
Hispanic	80	70	36	94

PPV: positive predictive value; NPV: negative predictive value.

## Data Availability

Data are contained within the article.

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
