# Peer review of "Racial Differences in the Detection Rate of Bladder Cancer Using Blue Light Cystoscopy: Insights from a Multicenter Registry"

_cancers, 2024, doi:10.3390/cancers16071268_

Round 1

Reviewer 1 Report

Comments and Suggestions for Authors

The study titled "Racial Difference in Detection Rate of Bladder Cancer Using Blue Light Cystoscopy: Insights from a Multicenter Registry" presents novel insights into the efficacy of Blue Light Cystoscopy (BLC) across different racial groups. Here's a summary of the main points, interest, novelty, and areas for improvement for potential publication.

Interest and Novelty:

  • Novel Insights on Racial Differences: The study fills a significant gap in the existing literature by investigating the detection rates of bladder cancer using BLC across various racial groups, a topic not extensively covered in previous studies.
  • Large Multi-center Data: Utilizing a comprehensive dataset from the Cysview® registry, spanning from 2014 to 2021 and including multiple US institutions, provides a robust basis for the findings, enhancing their generalizability.

Areas for Improvement:

  • Please Provide citation list. This is mandatory.
  • Distribution of CIS and Benign Lesions: The study should address the unequal distribution of CIS and benign lesions across racial groups more thoroughly, discussing how this might influence the interpretation of results and the generalizability of the findings.
  • Long-term Outcomes and Recurrence Rates: The absence of long-term follow-up data limits the ability to assess the impact of BLC on recurrence and survival rates. Please better describe this in the limitations
  • In a parallel advancement to the application of Blue Light Cystoscopy in enhancing bladder cancer detection, recent studies have also explored the utility of micro-ultrasound  especially for bladder cancers. According to a comprehensive review (PMID: 36363581), MUS has shown promising results in the high-sensitivity detection of clinically significant prostate cancer, comparing favorably with both systematic biopsy and mpMRI-guided biopsy in certain aspects. These innovations represent pivotal steps forward in tailoring patient management strategies, albeit with the acknowledgment that the evidence surrounding MUS is still preliminary and warrants further investigation. 
  • While the utilization of Blue Light Cystoscopy and Micro-Ultrasound marks a significant stride in the imaging-based detection and staging of urological cancers, emerging research on the urinary microbiome opens a new frontier in non-invasive cancer diagnostics. A recent study (PMID: 38298766) delves into the relationship between specific urinary bacteria and bladder cancer, shedding light on the potential of first-morning urine samples as a low-cost, accessible means for identifying individuals at increased risk of BCa. This research not only underscores the intricacies of the urobiome's role in carcinogenesis but also complements existing imaging techniques by offering a promising avenue for early biomarker-based diagnosis of bladder cancer. Together, these advances highlight a comprehensive approach to urological cancer care, blending the precision of imaging with the predictive power of molecular diagnostics. I strongly believe that those recent findings needs to be described and reported in your manuscript

Incorporating these improvements and emphasizing the study's contribution to the field could enhance its publication worthiness.

Author Response

The study titled "Racial Difference in Detection Rate of Bladder Cancer Using Blue Light Cystoscopy: Insights from a Multicenter Registry" presents novel insights into the efficacy of Blue Light Cystoscopy (BLC) across different racial groups. Here's a summary of the main points, interest, novelty, and areas for improvement for potential publication.

Interest and Novelty:

  • Novel Insights on Racial Differences: The study fills a significant gap in the existing literature by investigating the detection rates of bladder cancer using BLC across various racial groups, a topic not extensively covered in previous studies.
  • Large Multi-center Data: Utilizing a comprehensive dataset from the Cysview® registry, spanning from 2014 to 2021 and including multiple US institutions, provides a robust basis for the findings, enhancing their generalizability.

Areas for Improvement:

  • Please Provide citation list. This is mandatory.

Reply: Thanks much for bringing this up. The original manuscript submitted to the journal did include a reference list; however, it seems that this has been omitted when the file was sent to the reviewers. We re-included the citation list in the revised manuscript (pages 8 and 9).

  • Distribution of CIS and Benign Lesions: The study should address the unequal distribution of CIS and benign lesions across racial groups more thoroughly, discussing how this might influence the interpretation of results and the generalizability of the findings.

Reply: Thanks much for your comment. The details of CIS distribution have been presented in section 3.4 (lines 155-163) and Figure 1. More details regarding the prevalence of benign lesions have been added to the manuscript as well (lines 158-160). The effect of the unequal distribution of these lesions on the accuracy of blue light cystoscopy was highlighted in the discussion (lines 192-194 and 243-248). We re-emphasized this in the limitations (line 259).

  • Long-term Outcomes and Recurrence Rates: The absence of long-term follow-up data limits the ability to assess the impact of BLC on recurrence and survival rates. Please better describe this in the limitations.

Reply: We appreciate your comment. Nevertheless, reporting the long-term outcomes of patients is beyond the scope of this study. As the title suggests, the focus of this study is detection not recurrence/survival outcomes.

  • In a parallel advancement to the application of Blue Light Cystoscopy in enhancing bladder cancer detection, recent studies have also explored the utility of micro-ultrasound especially for bladder cancers. According to a comprehensive review (PMID: 36363581), MUS has shown promising results in the high-sensitivity detection of clinically significant prostate cancer, comparing favorably with both systematic biopsy and mpMRI-guided biopsy in certain aspects. These innovations represent pivotal steps forward in tailoring patient management strategies, albeit with the acknowledgment that the evidence surrounding MUS is still preliminary and warrants further investigation.

Reply: Thanks much for your comment. We agree that novel diagnostic modalities have the potential to optimize cancer management.

  • While the utilization of Blue Light Cystoscopy and Micro-Ultrasound marks a significant stride in the imaging-based detection and staging of urological cancers, emerging research on the urinary microbiome opens a new frontier in non-invasive cancer diagnostics. A recent study (PMID: 38298766) delves into the relationship between specific urinary bacteria and bladder cancer, shedding light on the potential of first-morning urine samples as a low-cost, accessible means for identifying individuals at increased risk of BCa. This research not only underscores the intricacies of the urobiome's role in carcinogenesis but also complements existing imaging techniques by offering a promising avenue for early biomarker-based diagnosis of bladder cancer. Together, these advances highlight a comprehensive approach to urological cancer care, blending the precision of imaging with the predictive power of molecular diagnostics. I strongly believe that those recent findings need to be described and reported in your manuscript.

Reply: Thanks much for your suggestion. Despite the importance of new imaging tools and blood/urine-based biomarkers in the diagnosis of bladder cancer, we believe that discussing these topics is beyond the scope of our manuscript.

Incorporating these improvements and emphasizing the study's contribution to the field could enhance its publication worthiness.

Reviewer 2 Report

Comments and Suggestions for Authors

In their multicenter study involving over 1200 patients, the authors compare and discuss the differing sensitivity of blue light cystoscopy (BLC) for detecting bladder cancer accross different races (Caucasian, African american, Asian, Hispanic). The sensitivity of BLC was higher than that of white light cystoscopy (WLC) in White/Caucasian and Asian populations, but similar in African American and Hispanic populations. Overall, BLC seemed to be superior to traditional WLC, but combining both methods could increase the sensitivity for malignant lesions by 10%, regardless of race. While the percentage of patients varied among ethnicities, ranging from 85% (Caucasian) to 4% (Hispanic), this study represents the first comparison for BLC efficacy among different races. In their discussion, the authors also high-lighted such limitations of the study. To enhance the manuscript, the following issues should be addressed: 1) Authors’ addresses not listed

2) Clarification of acronym on line 43 for improved reader comprehension

3) References not listed, therefore potential of perhaps too many self-citations not excluded

Author Response

In their multicenter study involving over 1200 patients, the authors compare and discuss the differing sensitivity of blue light cystoscopy (BLC) for detecting bladder cancer accross different races (Caucasian, African American, Asian, Hispanic). The sensitivity of BLC was higher than that of white light cystoscopy (WLC) in White/Caucasian and Asian populations, but similar in African American and Hispanic populations. Overall, BLC seemed to be superior to traditional WLC, but combining both methods could increase the sensitivity for malignant lesions by 10%, regardless of race. While the percentage of patients varied among ethnicities, ranging from 85% (Caucasian) to 4% (Hispanic), this study represents the first comparison for BLC efficacy among different races. In their discussion, the authors also high-lighted such limitations of the study. To enhance the manuscript, the following issues should be addressed:

1) Authors’ addresses not listed.

Reply: Thanks for your comment. All the addresses were added to the manuscript (page 1).

2) Clarification of acronym on line 43 for improved reader comprehension

Reply: Thank you for pointing this out. The abbreviation “MIBC” has been defined.

3) References not listed, therefore potential of perhaps too many self-citations not excluded

Reply: Thanks for your comment. We believe that the reference list may have been omitted when the file was sent to the reviewers. We re-included the citation list in the revised manuscript (pages 8 and 9).